Tumor-infiltrating CD8+ lymphocytes predict different clinical outcomes in organ- and non-organ-confined urothelial carcinoma of the bladder following radical cystectomy

Zhang Shiqiang zhangshq9@mail2.sysu.edu.cn shiqiangzhang@foxmail.com 1 2 4
Wang Jun 1
Zhang Xinyu 3
Zhou Fangjian zhoufj@sysucc.org.cn 2
1 Zhongshan School of Medicine, Sun Yat-sen University , Guangzhou , China
2 Department of Urology, Sun Yat-sen University Cancer Center , Guangzhou , China
3 Shenzhen Luohu Maternity and Child Healthcare Hospital , Shenzhen , China
4 Current affiliation:  The Seventh Affiliated Hospital of Sun Yat-sen University , Shenzhen , China
Tretiakova Maria
Electronic publication date: 2017 Oct 13
Publication date: 2017
Volume: 5
Electronic Location ID: e3921
Received 2017 Jul 26; Accepted 2017 Sep 22
Copyright: ©2017 Zhang et al.
Copyright year: 2017
Copyright holder: Zhang et al.
License: This is an open access article distributed under the terms of the Creative Commons Attribution License, which permits unrestricted use, distribution, reproduction and adaptation in any medium and for any purpose provided that it is properly attributed. For attribution, the original author(s), title, publication source (PeerJ) and either DOI or URL of the article must be cited.
License URL: https://creativecommons.org/licenses/by/4.0/

Keywords: CD8+ TILs, Urothelial carcinoma of the bladder, Organ-confined disease, Non-organ-confined disease

Funding: Sun Yat-Sen University Guangdong Province Innovative Research Team Program 2011Y035 from China This work was supported by the 985 Project Fund from Sun Yat-Sen University and the Guangdong Province Innovative Research Team Program 2011Y035 from China. The funders had no role in study design, data collection and analysis, decision to publish, or preparation of the manuscript.

==============================
Tumor-infiltrating lymphocytes (TILs) are associated with better clinical outcomes in many tumors. TILs represent a cell-mediated immune response against the carcinoma. CD8+ TILs are a crucial component of cell-mediated immunity. The significance of CD8+ TILs has not been reported respectively in organ- and non-organ-confined urothelial carcinoma (UC) of the bladder. We explored the prognostic value of CD8+ TILs in the two groups. The presence of CD8+ TILs was assessed by immunohistochemical staining of whole tissue sections from 75 organ and 51 non-organ-confined disease patients with long-term follow-up, and its correlation with clinicopathological features and overall survival (OS) was determined. The CD8+ TIL immunohistochemical staining score was 0 (<1%), 1 (≥1%), 2 (≥5%), or 3 (≥10%) based on the percentage of positively stained cells out of total cells. A patient was considered CD8 negative if the score was 0. There were no associations between CD8+ TILs and age, sex, nuclear grade, and adjuvant or neoadjuvant chemotherapy in organ- and non-organ-confined disease. The presence of CD8+ TILs was seen more frequently in pTa-1 than pT2 stage (p = 0.033) in organ-confined disease. No associations between CD8+ TILs and pT stage, pN stage were found in non-organ-confined disease. CD8+ TILs were associated with better OS (log-rank test, P = 0.036) in non-organ-confined disease, but with poorer OS (log-rank test, P = 0.040) in organ-confined disease by the Kaplan–Meier method. In multivariate analysis, CD8+ TILs were an independent favorable prognostic factor in non-organ-confined disease, but were an independent unfavorable prognostic factor in organ-confined disease. These results suggest that CD8+ TILs have clinically significant anti-tumor activity in non-organ-confined disease, but may have pro-tumor activity in organ-confined disease. Therefore, we should be cautious if CD8+ TILs are aimed to be exploited in the treatment of bladder cancer.

Introduction

Bladder cancer is the ninth leading cause of cancer death in both sexes, with an estimated 429,800 new cases and 165,100 deaths in 2012 worldwide (Torre et al., 2015; Ferlay et al., 2015). In the US, bladder cancer was the second most common genitourinary cancer, with projected 79,030 new cases and 16,870 related deaths in 2017 (Siegel, Miller & Jemal, 2017). In China, there were an estimated 80,500 new cases and 32,900 deaths from bladder cancer in 2015 (Chen et al., 2016).

Bladder cancer can be divided into organ-confined (pTa/pT1/carcinoma in situ [CIS]/pT2 NX/N0 M0) and non-organ-confined (pT3/T4N any M any or pT any N + M any or pT any N any M+) disease at cystectomy based on extravesical invasion (Boorjian et al., 2008). More than 90% of bladder cancers are urothelial carcinoma (UC) (Kaufman, Shipley & Feldman, 2009), of which one characteristic is the presence of high rates of somatic mutations (Cancer Genome Atlas Research Network, 2014). These mutations may produce an increased number of cancer neoantigens and enhance the ability of the host immune system to recognize tumor cells (Chen & Mellman, 2013; Sanmamed & Chen, 2014; Chen, 2014).

The presence of tumor-infiltrating lymphocytes (TILs) has been associated with favorable prognosis in several neoplasms (Pages et al., 2005; Galon et al., 2006; Laghi et al., 2009; Zhang et al., 2003; Wang et al., 2016; Mahmoud et al., 2011). Among the TILs, CD8+ TILs may play a central role in anti-tumor immunity. Non-organ-confined UC of the bladder has worse prognosis than organ-confined disease. However, only two reported studies have assessed the effect of CD8+ TILs in UC of the bladder following radical cystectomy, which suggest the presence of CD8+ TILs is associated with favorable prognosis. If CD8+ TILs have the similar role in the two group patients of organ- and non-organ confined disease, there hasn’t been reported.

In this study, we evaluated CD8+ TILs by immunohistochemistry (IHC) in organ- and non-organ-confined UC of the bladder following radical cystectomy, and explored the association with overall survival (OS).

Materials and Methods

Patients and tumor specimens

Archival pathology tumor blocks were obtained from the Department of Pathology of Sun Yat-sen University Cancer Center, Guangzhou, China. We retrieved 126 consecutive formalin-fixed, paraffin-embedded radical cystectomy specimens of UC of the bladder obtained between January 2000 and September 2009, including 51 non-organ-confined and 75 organ-confined disease. Representative tumor specimens were selected upon review of hematoxylin and eosin slides. We reviewed all sections to confirm the original diagnosis and staged them according to the 2010 American Joint Committee on Cancer TNM (tumor-node-metastasis) classification. The clinicopathological features were retrospectively reviewed from the patients’ medical records. The OS was determined from the date of surgery to the date of death or censored on the date of the last follow-up. Written informed consent was obtained from all patients prior to the study. The Ethical Committee of Sun Yat-sen University Cancer Center approved all experimental methods in the study (YB2016-006).

CD8 IHC

CD8 IHC was conducted according to a standard method. Briefly, tissue sections were deparaffinized and rehydrated. Endogenous peroxidase activity was blocked with 3% hydrogen peroxide for 15 min. For antigen retrieval, tissue slides were boiled in 10 mM citrate buffer (pH 6.0) in antigen retriever (Ascend Biotechnology, Guangzhou, China) at 120 °C for 5 min. Nonspecific binding was blocked with Serotec Block ACE (AbD Serotec, Oxford, UK) for 15 min. The slides were incubated with anti-CD8 antibody (Clone SP16; Thermo Scientific, Fremont, USA) overnight at 4 °C. All incubations were performed in a moist chamber. Subsequently, the slides were incubated with a horseradish peroxidase–labeled secondary antibody for 60 min at 37 °C, and then visualized using 3,3′-diaminobenzidine.

Scoring of CD8+ TILs

Two independent pathologists blinded to the clinicopathologic information performed the scoring. The CD8+ TIL score was 0 (<1%), 1 (≥1%), 2 (≥5%), or 3 (≥10%) based on the estimated percentage of positively stained cells out of total cells. A patient with a score of 0 was considered CD8 negative (CD8−). If there was a discrepancy in scoring, both pathologists reviewed the case using a double-headed microscope to achieve consensus. About 31% of the total cases required a consensus discussion between the reviewing pathologists for scoring. At all times, the pathologists were blinded to the clinical outcome.

Statistical methods

The Fisher exact test was used to evaluate CD8 status between different categories. The Kaplan–Meier method and Cox proportional hazards model were used for OS analysis; differences in the Kaplan–Meier survival analysis were assessed using the log-rank test. Statistical analysis was performed using IBM SPSS 19.0 (SPSS, Inc., Chicago, IL, USA) for Windows; the significance level was set at 0.05 (2-sided).

Results

Clinicopathological analysis of organ and non-organ-confined disease and presence of CD8+ TILs

Table 1 summarizes the clinicopathologic characteristics. The median age of the 51 patients at surgical resection was 63 years, and the median follow-up time from date of surgical resection was 30.5 months (range: 0.2–129.6 months) in non-organ-confined disease. In organ-confined disease, median age of the 75 patients at surgical resection was 60 years, and median follow-up time from date of surgical resection was 51.8 months (range: 0.3–139.7 months).

Table 1 Correlation between clinicopathological features and CD8+ TILs.

Characteristics	N	Organ-confined	P	Characteristics	N	Non-organ-confined	P	
		CD8+	CD8−	Fisher exact test			CD8+	CD8−	Fisher exact test	
Age (years)				0.296					0.760	
<60	36	24	12			18	13	5		
≥60	39	31	8			33	22	11		
Sex				1.000					0.295	
Male	64	47	17			47	31	16		
Female	11	8	3			4	4	0		
pT stage				0.033	pT stage				0.345	
pTa-1	31	27	4		pT3	34	25	9		
pT2	44	28	16		pT4	17	10	7		
pN stage					pN stage				0.746	
					N−	35	23	12		
					N+	16	12	4		
Nuclear grade				0.107						
Low	26	16	10							
High	49	39	10							
Adjuvant chemotherapy, n (%)				0.114					0.527	
Not Administered	70	53	17			34	22	12		
Administered	5	2	3			17	13	4		
Neo-adjuvant chemotherapy, n (%)				1.000					0.643	
Not Administered	65	48	17			5	3	2		
Administered	10	7	3			46	32	14		

CD8+ TILs were detected in both the tumoral and intratumoral areas, and were distributed mainly at the interface of the tumor and tumor-adjacent stroma (Fig. 1). We compared CD8+ TILs with the clinicopathological characteristics of organ- and non-organ-confined disease (Table 1). There were no associations between CD8+ TILs and age, sex, nuclear grade, and adjuvant or neoadjuvant chemotherapy in both organ- and non-organ-confined disease. In organ-confined disease, the presence of CD8 TILs was seen more frequently in pTa-1 than pT2 stage (p = 0.033). No associations between CD8+ TILs and pT stage, pN stage were found in non-organ-confined disease.

OS and Cox proportional hazards analysis

We examined the relationship between the presence of CD8+ TILs and OS from the date of cystectomy. There was no statistically significant association between CD8+ TILs and OS in all 126 UC of the bladder patients (OS, P = 0.899, Fig. 2A) by the Kaplan–Meier method. The presence of CD8+ TILs was correlated with improved OS in 51 patients of non-organ-confined disease (P = 0.036, Fig. 2B). On the contrast, it was associated with poorer OS in 75 patients of organ-confined disease (P = 0.040, Fig. 2C). The multivariate analysis showed that the presence of CD8+ TILs was an independent prognostic predictor in both organ- and non-organ-confined disease. In organ-confined disease, the patients with CD8 negative had a better prognosis than the patients with CD8 positive (RR: 0.212, 95% CI [0.048–0.932], P = 0.0040; Table 2). However, in non-organ-confined disease, the patients with CD8 negative had a worse prognosis than the patients with CD8 positive (RR: 2,397, 95% CI [1.031–5.573], P = 0.042; Table 2).

Figure 1 Representative examples of CD8+ TILs immunostaining.

CD8+ TILs are distributed mainly at the interface of the tumor and tumor-adjacent stroma. (A), (B), (C) and (D) (×100 magnification) are representative images of scores of 0, 1, 2, and 3, respectively.

Figure 2 Kaplan–Meier survival curves with log-rank testing.

(A) All UC of the bladder (CD8+, 90; CD8−, 36). (B) Non–organ-confined UC of the bladder (CD8+, 35; CD8−, 16). (C) Organ-confined UC of the bladder (CD8+, 55; CD8−, 20).

In the present study, the CD8+ TIL score was 0 (<1%), 1 (≥1%), 2 (≥5%), or 3 (≥10%) based on the estimated percentage of positively stained cells out of total cells. It was no statistically significant association between CD8+ TILs and OS in all 126 UC of the bladder patients by the Kaplan–Meier method regardless of CD8 score = 1 & 2 & 3 vs CD8 score = 0, CD8 score = 0 & 1 vs CD8 score = 2 & 3). There was no statistically significant association in non-organ-confined disease or pT3 + 4 stage (data not shown) when it was CD8 score = 0 & 1 vs CD8 score = 2 & 3; but when it was CD8 score = 1 & 2 & 3 vs CD8 score = 0, there was statistically significant association in non-organ-confined disease or pT3 + 4 stage (data not shown).

Discussion

We show that CD8+ TILs are significantly associated with favorable clinical outcome in non-organ-confined disease, but with unfavorable clinical outcome in organ-confined disease in the Kaplan–Meier survival analysis, although no statistically significant association between CD8+ TILs and OS was found in all 126 UC of bladder. In multivariate Cox’s proportional hazards regression analysis, the presence of CD8+ TILs was an independent factor of prognosis in both of organ-confined and non-organ confined disease. It was a favorable factor of prognosis in non-organ-confined disease, but was an unfavorable factor in organ-confined disease.

Previous studies have shown that TILs (especially CD8+ TILs), which have anti-tumor activity, are associated with good prognosis in patients with tumors (Galon et al., 2006; Mahmoud et al., 2011; Hwang et al., 2012; Pages et al., 2005). CD8+ TILs mediate most anti-tumor immune responses (Wang et al., 2016), bladder cancer studies also suggest that the presence of CD8+ TILs is associated with favorable prognosis (Sharma et al., 2007; Faraj et al., 2015). High CD8 density was associated with improved OS (P = 0.02) in a subset of 50 cystectomies for invasive bladder UC (≥pT1) (Faraj et al., 2015). The authors defined high CD8 density as the presence of ≥60 CD8+ T cells per high-power field in a given tissue microarray spot. A tumor was considered high density if 50% of its spots were of high density. This is different from our study. A patient with a score of 0 (<1% of positively stained cells out of total cells ) was considered CD8 negative in our study. CD8+ T cells were detected in both the tumoral and intratumoral areas, and were distributed mainly at the interface of the tumor and tumor-adjacent stroma (Tumeh et al., 2014), the score of CD8+ TILs was accessed in tumor periphery (i.e., the interface of the tumor and tumor-adjacent stroma) in the present study. Another study showed that, in similar-staged bladder UC (pT2, pT3, or pT4), patients with more CD8+ TILs within the tumor had better OS (P < 0.018) than patients with fewer CD8+ TILs (Sharma et al., 2007). Multivariate analysis also revealed a significant association between CD8+ TILs and OS. The previous two reports had the similar results, and it was consistent with the findings of the present study in non-organ confined disease. However, the presence of CD8+ TILs is an significant unfavorable predictor in organ-confined disease, which was also found in the present study. The immune cells of CD8+ TILs have anti-tumor activity in non-organ confined disease, and may have pro-tumor activity in organ-confined disease. But if the CD8+ TILs themselves had pro-tumor activity or CD8+ TILs’ anti-tumor activity inhibited by other powerful pro-tumor factors in organ-confined disease, such as tumor cells PD-L1 expression (Chen & Flies, 2013), it needs further investigation. Many studies suggest CD8+ TILs could potentially be exploited in the treatment of cancer. If CD8+ TILs are aimed to be exploited in the treatment of bladder cancer, that different therapeutic strategies should be considered, which was indicated by the present study.

Table 2 Association between CD8 status and OS in the subset of 75 organ-confined and 51 non-organ-confined urothelia carcinoma of the bladder patients in the Cox proportional hazards model.

	Organ-confined	Non-organ-confined	
	Risk ratio	95 % CI	P	Risk ratio	95 % CI	P	
CD8− vs CD8+	0.212	0.048–0.932	0.040	2.397	1.031–5.573	0.042	
Notes.

Adjusted for age, sex, nuclear stage, T stage, N stage (only in non-organ-confined), neo-adjuvant chemotherapy, and adjuvant chemotherapy.

The efficacy of immunotherapy in superficial bladder tumors was first established in 1976 with BCG (Morales, Eidinger & Bruce, 1976), but no immunotherapy had been approved for the treatment of advanced disease, until anti-PD-1/PD-L1 monoclonal antibody were approved for the treatment of bladder cancer patients (Powles et al., 2017). All of the patients in our study didn’t receive any immunotherapy.

The study limitations are the small number of patients included, the heterogeneity of the patient population and contrasting results between organ- and non-organ-confined disease. Therefore, further investigation is necessary to verify our findings, and explore why the presence CD8+ TILs have different effects in the two cohorts.

In conclusion, our results provide evidence of the prognostic importance of CD8+ TILs in organ- and non-organ-confined UC of the bladder. The presence of CD8+ TILs in non-organ-confined disease correlates with improved OS, but it was correlated with poorer OS in organ-confined disease. On this account, if CD8+ TILs are aimed to be exploited in the treatment of bladder cancer, different treatment strategies should be considered.

Supplemental Information

Data S1 Raw data

Click here for additional data file.

We thank Prof. Lieping Chen (Yale University) for experimental design and laboratory assistance. We thank Xinke Zhang (Sun Yat-sen University Cancer Center) and Weisong Li (The First Affiliated Hospital of Sun Yat-sen University) for their assistance with the scoring of CD8+ TILs.

Additional Information and Declarations

Competing Interests

Author Contributions

Human Ethics

Data Availability

The authors declare there are no competing interests.

Shiqiang Zhang conceived and designed the experiments, performed the experiments, analyzed the data, contributed reagents/materials/analysis tools, wrote the paper, prepared figures and/or tables, reviewed drafts of the paper.

Jun Wang performed the experiments, contributed reagents/materials/analysis tools, wrote the paper, reviewed drafts of the paper.

Xinyu Zhang analyzed the data, contributed reagents/materials/analysis tools, wrote the paper, prepared figures and/or tables, reviewed drafts of the paper.

Fangjian Zhou conceived and designed the experiments, wrote the paper, reviewed drafts of the paper.

The following information was supplied relating to ethical approvals (i.e., approving body and any reference numbers):

The Ethical Committee of Sun Yat-sen University Cancer Center approved all experimental methods in the study.

The following information was supplied regarding data availability:

The raw data has been uploaded as Data S1.

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
