# Peer review of "Tumor-infiltrating CD8+ lymphocytes predict different clinical outcomes in organ- and non-organ-confined urothelial carcinoma of the bladder following radical cystectomy"

_PeerJ, doi:10.7717/peerj.3921_

## Round 0.1 · original submission · Minor Revisions

· Academic Editor

Minor Revisions

There are minor, but very numerous deficiencies of this manuscript that will require detailed response to both reviewers. As an Editor, I also have some comments that needs to be addressed:
1) No information was provided regarding BCG therapy which is known to elicit increase in TILs and should be correlated with CD8 positivity.
2) The scoring of CD8 TILs should be further clarified to whether it was accessed in tumor center or periphery, and what was the significance of generating 4-tier scoring system.
3) TCC is an outdated term and should be replaced with Urothelial carcinoma (UC) throughout the text.
4) There are typos in the table.

Reviewer 1 ·

Basic reporting

Language used:
The language should be edited for clarity.
Sentences that are unclear:
Lines 64-65: please edit for clarity/make more concise
Lines 152-154: Unclear if “the authors” defining high CD8 density are the authors of the present study or if this sentence refers to the most recently referenced papers; please edit for clarity and insert reference(s) as appropriate.
Lines 163-165: This sentence invokes possible (and likely very important) immunosuppressive effects of the local tumor microenvironment (TME), but is unclear; please edit and insert reference(s) as appropriate.
Lines 174: As is, this is a long sentence; suggest refining to minimize redundant statements.

Sentences that could be improved for style and readability:
Line 148-150: awkward construction of final clauses
Lines 158-160: awkward construction
Line 175-176: wordy

Literature references, etc:
No comments.

Article Structure:
Lines 132-135: The statement describing the results of the multivariate Cox proportional hazards regression model is closer to a conclusion than a simple statement of findings; please focus on describing the correlation identified without straying into interpretation (value as prognostic/predictive factor) in the results section.

Self-contained with relevant results to hypothesis:
See comments below (Experimental Design section) for recommended additional analyses to include in the manuscript; several may have already been performed but should be explicitly stated.

Experimental design

Aim & scope:
Appropriate.

Research Question:
The study adds to the literature by demonstrating differential correlation between overall survival (OS) and infiltrating CD8+ T-cells in organ-confined and non-organ confined disease. OS is a strong, important end-point while the semi-quantitative method for scoring CD8 positive cells infiltrating into neoplasm is likely reproducible in the clinical setting.

Rigorous Investigation:
It would be interesting if the authors examined correlation of CD8+ TILs and OS for pT2-4 lesions as a comparison to Sharma et al (PMID: 17360461); this would also test the hypothesis that the negative correlation between CD8+ TILs and OS in organ confined disease is driven primarily by pT1a lesions.


Method description:
1) The authors should clarify their scoring methods – is the percentage CD8+ cells out of total cells or total lymphocytes? Was this an estimate or was this counted? An estimate is acceptable, but should be stated explicitly.

2) The authors should explicitly discuss in the manuscript why/how they selected bins for CD8+ TILs in their analysis (I.e., CD8+ vs CD8-negative, as opposed to other score combinations such as CD8+ score = 0 & 1 vs CD+ score = 3 & 4)

3) If known/recorded, it would be nice to include how many cases required a consensus discussion between the reviewing pathologists for scoring.

Validity of the findings

Data
Robust; details provided. Concerns are expressed above about method description and other possible analyses that might make the results more impactful and demonstrate both consistency with prior literature and highlight new findings (i.e., differential correlation of CD8 TILs with OS in low-stage vs high-stage disease).

Conclusions
Conclusions would benefit from increased clarity in writing and more concise statements.
A minor point: the authors have not demonstrated activity (pro or anti-tumor) of the TILs. They have demonstrated presence of CD8+ cells alone; activity would require markers of T-cell activation. In general this distinction is clear, but grows slightly fuzzy in lines 162-5. Afterall, as the authors note in Lines 164-5, the local tumor microenvironment very likely has an important role.

Speculation
No comment

Additional comments

No additional comments

·

Basic reporting

The manuscript meets standards. Minor points and comments are as follows:
1. Row 48: These 2017 numbers are projected and not actual numbers. Please revise.
2. Did any of the patients receive any immunotherapy. A comment on the existing knowledge on immunotherapy of TCCs could be useful.
3. Rows 142-143: The authors write “…however, which was a favorable…”. Please rephrase for syntax.

Experimental design

The manuscript meets standards.

Validity of the findings

The authors investigate the presence of CD8-positive tumor infiltrating lymphocytes (TILs) in organ-confined and non-organ-confined excision specimens of bladder transitional cell carcinomas. Based on their findings, the authors conclude that the presence of TILs is an independent prognostic factor in both organ-confined (unfavorable) and non-organ-confined (favorable) disease. No other clinicopathological correlations are identified based on the findings of the present study. While technically statistically significant in both cohorts (p<0.005), the statistical correlation is not compelling, as the reported p values are approaching the 0.005 cutoff. To further weaken the findings, the results are contrasting between the two cohorts. The authors should state this as a limitation of this study and should also try to offer possible explanations to the contrasting results between organ-confined and non-organ-confined disease. Finally, some information should be offered, with regard to the current state of immunotherapy in bladder transitional cell carcinoma, as this closely relates to the presence of TILs. Although unlikely, information (or lack thereof) should be included on whether patients have had immunotherapy as part of their treatment, and report any correlations with presence of CD8+ TILs.

---

## Round 0.2 · Minor Revisions

· Academic Editor

Minor Revisions

Although the rebuttal and revisions were substantial and significantly improved manuscript quality, some of criticisms from reviewer 1 were not adequately addressed. Therefore, we urge you to complete 4 minor albeit important, requested modifications (see below).

Reviewer 1 ·

Basic reporting

This is a re-review of the manuscript submitting following minor requested revisions. I have four truly minor, albeit important, requested modifications.
1) The authors have now twice addressed concerns expressed from multiple reviewers about the selection of bins of CD8+ scores and explained their rationale for using CD8+ TIL negative vs positive (grouping 0 & 1 scores vs 3 & 4 scores). They have adequately explained in the rebuttal letters that there was no statistically significant association for other combinations of scores. However, unless I missed something, the authors have still not included this explanation in the primary text. I strongly believe this rationale should be explicitly discussed in the text (the results section would be appropriate). Without such a clear, concise discussion in the results section: a) the rationale behind Figure 2's construction is unclear; and b) the readership is left with outstanding methodological questions. The single sentence provided in the rebuttal letter is sufficient explanation and an important result.
2) Similarly, the authors explain in the rebuttal letter that percentage of CD8+ TILs was estimated as a percentage of total cells. That methodology is definitely acceptable (to me, at least). However, this has not been made explicit in the manuscript text. I suggest modifying Lines 89-90 as follows: "based on the [estimated] percentage of positively stained cells out of total cells." (Adding "estimated" to existing text; brackets indicate my modification/suggestion).
3) Figure 2 still uses TCC (transitional cell carcinoma) in lieu of urothelial carcinoma (UC) which has replaced all other uses of TCC in the text.
4) A minor point: there is inconsistent use of "Figure X" vs "Fig X"; please compare lines 109 and 123 and choose one for consistent style.

Experimental design

No concerns.

Validity of the findings

No concerns.

Additional comments

I believe this paper is deserving of publication and congratulate the authors on their efforts. However, I believe the four really minor corrections in Part 1 (above), particularly the first two points, really are necessary corrections prior to publication and will better enable colleagues' interpretation of the data and results, and the TIL community's ability to replicate the methods in the paper.

·

Basic reporting

No comment.

Experimental design

No comment.

Validity of the findings

No comment.

Additional comments

The authors have adequately addressed the reviewers' comments and have presented an improved version of this manuscript

---

## Round 0.3 · accepted · Accept

· Academic Editor

Accept

While in production, please double check the English in your corrected sentences. There are some minor imperfections.